# Exploring the Potential of Big Data Analytics in Urban Epidemiology Control: A Comprehensive Study Using CiteSpace

**DOI:** 10.3390/ijerph20053930

**Published:** 2023-02-22

**Authors:** Jun Liu, Shuang Lai, Ayesha Akram Rai, Abual Hassan, Ray Tahir Mushtaq

**Affiliations:** 1School of Mechanical Engineering, Northwestern Polytechnical University, Xi’an 710072, China; 2School of Public Policy and Administration, Northwestern Polytechnical University, Xi’an 710072, China; 3School of Medicine, Xi’an Jiaotong University, Xi’an 710049, China; 4Faculty of Mechanical Engineering and Ship Technology, Gdansk University of Technology, 80-233 Gdansk, Poland

**Keywords:** big data, pandemic control, urban health, sustainability, energy, privacy

## Abstract

In recent years, there has been a growing amount of discussion on the use of big data to prevent and treat pandemics. The current research aimed to use CiteSpace (CS) visual analysis to uncover research and development trends, to help academics decide on future research and to create a framework for enterprises and organizations in order to plan for the growth of big data-based epidemic control. First, a total of 202 original papers were retrieved from Web of Science (WOS) using a complete list and analyzed using CS scientometric software. The CS parameters included the date range (from 2011 to 2022, a 1-year slice for co-authorship as well as for the co-accordance assessment), visualization (to show the fully integrated networks), specific selection criteria (the top 20 percent), node form (author, institution, region, reference cited, referred author, journal, and keywords), and pruning (pathfinder, slicing network). Lastly, the correlation of data was explored and the findings of the visualization analysis of big data pandemic control research were presented. According to the findings, “COVID-19 infection” was the hottest cluster with 31 references in 2020, while “Internet of things (IoT) platform and unified health algorithm” was the emerging research topic with 15 citations. “Influenza, internet, China, human mobility, and province” were the emerging keywords in the year 2021–2022 with strength of 1.61 to 1.2. The Chinese Academy of Sciences was the top institution, which collaborated with 15 other organizations. Qadri and Wilson were the top authors in this field. The Lancet journal accepted the most papers in this field, while the United States, China, and Europe accounted for the bulk of articles in this research. The research showed how big data may help us to better understand and control pandemics.

## 1. Introduction

Owing to rising instances of natural catastrophes, health pandemics, political uprisings, and gang violence, urban cities have become the “locus of hazards” [1]. Despite their economic success and their standing as centers of cultural interchange and technological innovation, megacities—which the United Nations classifies as having more than 10 million residents—present rising hazards to ecological and population health. There will be 43 giant cities by 2030, according to the United Nations, and most of them will be in developing nations. There will be approximately 10 billion people on Earth by 2050, with approximately 68% residing in cities [2]. Megacities are especially susceptible to the spread of infectious illnesses, including dengue and acute respiratory syndrome (SARS syndrome) because of the high population density and ease of travel across neighborhoods. Meanwhile, advances in fields such as ICT, the Internet Of Things, cloud technology, and smartphone applications have made it possible for people to share information in near real time. A more in-depth and all-encompassing comprehension of urban circumstances and real-time events is made possible by the huge volume, velocity, and diversity of urban data [3].

The current coronavirus illness (COVID-19) epidemic has captured the attention of the whole world as it has unfolded. Wuhan, a Chinese megalopolis of around 11 million people, was the first to report the epidemic [4]. The medical community’s attention was instantly drawn to the epidemic as the number of verified cases and fatalities skyrocketed. While there is a richness of study on the environmental, psychological, economic, and health effects of urban epidemics, the vast majority of such literature is devoted to strategic planning and sustainable policy analysis. Epidemic dynamics (e.g., prospective transmission and infection patterns) between cities have been shown to vary due to endogenous factors at the city level, including geography, demographic characteristics, spatial organization, regional connectedness, and microclimate [5].

Artificial intelligence (AI) and big data, enabled by recent developments in computational methods and information and communications technology (ICT), can assist in managing the massive, unprecedented volume of data resulting from surveillance, real-time monitoring of epidemic outbreaks, trend now-casting/forecasting, regular circumstance briefing and updating from government institutions and health organizations, and health resource utilization data [6]. 

Classical definitions of big data focus on the three characteristics of “velocity”, “volume”, and “variety”, all of which speak to the unprecedented speed with which data may be acquired, processed, and manipulated that characterizes big data (also known as “fast data”) [7]. Big data can be broken down into many different categories based on its sources, including: (i) molecular big data, which are obtained through lab methods and post-genomic specialties, such as proteomics and interactomics; (ii) optics big data, which includes radiomics or the large data method to extract useful clinical, elevated information from images; (iii) device big data, which includes wearable sensors; and (iv) digital and data-processing big data [8,9].

Tremendous work on the use of big data to control pandemics has been achieved. However, the scientometric analysis of this topic has not yet been accomplished. Using the bibliometric tool CiteSpace (CS), we wanted to give a thorough and up-to-date assessment of the literature on the subject of big data-based urban epidemic control. When it comes to visualizing and analyzing vast and complicated collections of bibliographic data, CS is an invaluable tool [10,11]. The method has seen extensive application in epidemiology, among other fields, because of its ability to spot repeating patterns in published research. The bibliometric analysis for this research was based on data collected from Web of Science (WOS) and spanned 11 years in order to include the most up-to-date and relevant publications in the field. To identify major authors, significant publications, and developing trends in the area of urban outbreak control based on big data, the research employed specified keywords and criteria to extract relevant literature, with CS used for display and assessment of the data [12,13,14]. 

Our research employed CS as a research tool to assess co-authorship, co-citations, and keywords from foreign studies. The following steps were taken to achieve the CS objective:I.Retrieval and assessment of the major big data pandemic control-based research, pinpointing the authors, institutions and countries, so as to;II.Identify where the primary publications that were crucial to the study were published, in order to;III.Categorize the primary research concept and explore the knowledge architecture in this field, and to;IV.Locate active research areas and new horizons in the field of epidemic control based on big data research.

## 2. Methodology

### 2.1. Web of Science (WOS) Usability

The WOS is frequently regarded as the world’s best source of data for bibliometric analysis [15,16]. Thus, Web of Science Core Collection (WOSCC) data served as the study’s data sources. On 30 December 2022, the data in this study were collected for a time frame of “11 years”. To find the most pertinent publications in the area of big data-controlled pandemics, numerous keywords were considered. Included below are some of the most often used keyword codes for executing searches in order to retrieve information from WOS databases, as shown in Table 1. The following is a list of the most productive search keywords: ((“endemic” OR “pandemic”) AND (“city” OR “Urban”) AND (“big data” or “electronic health records” OR “social media posts” OR “sensor data” OR “Artificial intelligence” OR “AI” OR “Machine learning”) AND (“control”)) which specified that original available articles [17] containing such words located in abstracts, keywords, or titles, were downloaded. Initially, the authors found 243 articles, of which 223 were original articles (including early access), 2 were proceedings papers, and 15 were review articles, as well as 1 letter, 1 meeting abstract, and 1 editorial piece. Then, authors manually excluded the review papers, proceedings, letter, meeting abstract and editorial piece to enhance the quality of bibliometric analysis, leaving 223 articles selected for further analysis. The authors then restricted the time frame to 2011–2022 and found 208 original articles. Early-access articles were then excluded, leaving 202 articles for further analysis.

### 2.2. CiteSpace Visualization

Among bibliometric approaches, social network analysis study visualization is a developing field. Nine significant software programs have been specifically created to use research mapping to assess scientific fields [18]. Chaomei Chen developed the CS, an open Java science research software package from Drexel University in the United States [19]. Owing to its strong and effective properties, it is widely used and recognized throughout the world [20]. As a result, CS was used to visualize and evaluate epidemic control based on big data research literature. 

The main analytical processes were used to visually assess and collect the data through the CS (6.1.R6). First, a project proposal for a project titled “Big data technologies to control pandemic” was created using the WOS’ exhaustive list and quoted a straightforward CS text tutorial. The authors used only WOSCC due to the CS software’s limitation of only being able to use one kind of research collection at one time; the authors tried to use other research collections along with WOSCC, but CS could not visualize the data. The second step was to clarify the parameters, which included the date range (from 2011 to 2022, a 1-year slice for co-authorship as well as for the co-accordance evaluation), visualization (displaying the merged networks), specific selection criteria (top 10 percent), node form (writer, institution, region, credible source cited, cited writer, journal, and keywords), pruning (including the pathfinder as well as the slicing network), and specific selection process (top 10 percent). Figure 1 shows the CS parameter selection to visualize the data derived from WOS. 

The CS was operated by acquiring a network and data highlighting authors, institutions, and countries that had published more than two articles; journals; and authors who were cited twice or more, in addition to co-author, co-citation, and evaluation of co-occurrences with 1 or more occurrences. The outcomes of this research’s visualization study were researched and displayed by correlating data and networks. 

The current CS-based study pinpoints the most important research areas, hotspots, and frontiers in big data pandemic control schemes research to advanced big data technology in a variety of applications; consequently, it may be very beneficial for future research. 

## 3. Big Data Applications in COVID-19

Big data can offer tremendous potential for controlling COVID-19 and other emergencies, and its role is projected to increase in the future. AI and big data are used to monitor the transmission of infection in real time, strategize and improve health programs accordingly, monitor the efficacy of these initiatives, repurpose old compounds, discover new drugs, identify possible vaccine candidates, and enhance the reaction by groups and regions to an ongoing pandemic. These new technologies may be used in concert with conventional surveillance to offer even more accurate forecasts, as big data can enable interpretation and analysis while AI can expose previously overlooked trends and patterns.

A comprehensive and in-depth investigation was done by Li et al. [21] to determine the best methods for detecting and eliminating erroneous reports of epidemics and for forewarning the public about such reports. This article offered a fresh viewpoint on the study of methods for preventing and resolving epidemics by examining the possibility of doing so using big data on care and wellness.

An application of the IoMT big data platform, cov-AID, was introduced by Hamid et al. [22] to help stop the spread of COVID-19. The suggested system enables the identification of hotspots for the spread of COVID-19, allowing for the monitoring and treatment of patients at a distance. Remote diagnosis, continuous monitoring, speedy treatment in the convenience of one’s own home, and protection against the spread of the virus are all areas where the cov-AID architecture shows great promise. Patients in emergency situations of may be handled quickly and cheaply with the help of the fast strategies. The cov-AID platform reliably stores massive datasets for illness forecasting and online consultations at a distance.

Levashenko [23] presented a novel approach to structure function building for the mathematical expression of a system that relies on incompletely described and uncertain data, with potential applications across a range of healthcare-related systems. Additionally, a technique for constructing such a structure was provided by the authors. In a resilience engineering context, constructing a mathematical model of the system under study was a crucial first step, since it facilitated the use of techniques for determining a system’s dependability based on a variety of different indices and metrics. When it came to reliability engineering, the structuring function’s capacity to clearly represent a system of any structural complexity and its wide availability of well-developed techniques for the computation of various metrics were two of the representation’s most valuable features. The suggested technique was based on classifier induction, which was used to generate the structure function, and the computation of the case data as the classification problem.

The authors [24] examined in depth the potential of combining big data and AI for use in classroom settings. During the period of the New Crown pandemic, it was important to learn how to deal with the pressures of daily living. Use of the internet, large datasets, and AI were the key tools used to solve the problem. Computer-aided instructional design, multimedia teaching design, and informative instructional design were examined to see whether they prompted student migration using the paradigm limited shift analysis framework. The paper proposed three solutions to the problem: more awareness, improved planning, and more robust exercises.

During the COVID-19 epidemic, tourist services were improved via the use of information and communication technology. The goal of the research was to find ways that the tourist sector may use information technology to improve service quality despite the limitations imposed by COVID-19. It is not a coincidence that virtual reality is becoming more popular and that guided online tours are the method of choice for professionals throughout the globe. As a result of the COVID-19 epidemic and the subsequent limitations, the tourist industry and its activities were shut down. The tourist sector as a whole will feel the effects of the epidemic for a long time after it has finished [25].

Wenyu Chen and his colleagues [26] looked at big data technology architecture during the COVID-19 epidemic in China. As of 20 January 2020, the Coronavirus illness 2019 in China was classified as a Class B contagious disease but was managed in line with Class A contagious diseases due to its high contagiousness and fast dissemination. In their study, they detailed the design and implementation processes, as well as some of the early results of, a novel big data management system for evaluating COVID-19 personnel. During the early phases of the pandemic, one of the parameters used to determine a patient’s risk of contracting COVID-19 was the health Qr, a big data technology platform created in China. By the end of the year 2021, over 690,000 persons will have used automated regulation for protection and monitoring, equating to around 5.79 million person-hours.

Hua and Shaw [27] argued that the People’s Republic of China’s success in containing COVID-19 could be attributed in large part to “a unique combination of advanced governance, strict regulation, great community surveillance and civic engagement, and sensible use of big data technology and digital technologies” despite an initial delay in response from the Chinese authorities. More study is needed to determine the best way to use such complex technology while protecting people’s privacy and remaining true to ethical principles. Possible uses of big data for the management of an ongoing COVID-19 outbreak are depicted in Table 2 [3]. 

As the COVID-19 epidemic swept the globe, it generated a massive and ever-expanding trove of data. Big data analytics approaches may be used with this information to aid in a variety of settings, such as diagnostics; risk score estimation and prediction; healthcare decision-making; and the pharmaceutical business. Figure 2 shows several possible uses of big data in COVID-19 [28].

## 4. Results and Discussion on CiteSpace Visualization

CiteSpace is a software tool for visualizing and analyzing scientific literature data. It is commonly used in scientometrics, which is a field of study that uses bibliometric methods to measure and analyze scientific and technological research.

CiteSpace provides a number of features that support the analysis of scientific literature, including identification of key researchers, institutions, and journals in a particular field, as well as analysis of co-citation networks to identify influential articles and intellectual bases of research fields. CiteSpace can be used to identify trends and patterns in scientific research, to track the development of a particular field, and to identify areas for future research. The software is flexible and can be applied to a wide range of scientific and technological fields. For pandemic control using big data, the following is the extensive context for the whole investigation, as shown in Figure 3.

### 4.1. Co-Authorship Analysis of Studies of Epidemic Control Based on Big Data Research 

The most effective and well-acknowledged example of scientific collaboration is the management of epidemics via the analysis of large datasets; this kind of work involves collaboration on national, personal, and institutional levels. The study of co-authorship in epidemic control based on big data research is useful for learning about the working relationships between several authors and for finding active epidemic control based on big data researchers from all over the globe [29]. 

#### 4.1.1. Author–Co-Authorship Analysis of Studies of Epidemic Control Based on Big Data Research 

Figure 4 represents the scholarly efforts of several writers working together to generate a single piece of work by deciding on an appropriate threshold and an appropriate analytical unit. The size of a node represents the sum of all articles written by its writers. Links between writers are thicker when they work together. The colors represent the publishing year, ranging from blue (oldest papers) to yellow (most recent papers). As seen in Figure 4, the authors of at least one article were graded. The most renowned scholarly team under Qadri Firdausi’s [30] supervision is a large sub-network with six nodes, while Chowdhury Fahima [31], Cravioto Alejandro [32], and Khan Ashraful Islam [33] were also incorporated in his study team. In one of their papers [30], oral cholera vaccination was described by Qadri and colleagues as both feasible and efficacious in an urban endemic context in Bangladesh. An annual epidemic of cholera is a fact of life in Bangladesh. The decision on whether or not a low-cost, oral killed whole-cell cholera vaccine is used to manage the illness in a public health environment relies on the vaccine’s acceptability and effectiveness in that context. Anti-harming efficacy among vaccinated people was 37% overall. Dhaka, Bangladesh is home to several urban slum communities, and another piece of research [31] was undertaken there. One third of the population lived in “better” WASH homes, and the occupants had a 47% lower risk of severe cholera. In terms of severity, cholera risk was most strongly connected with poor water quality and limited access to clean water. Living in a household with “superior” WASH facilities was associated with lower mortality across all age categories. Dhaka, Bangladesh, is home to several urban slum communities, where our research took place. The onset of the first severe cholera episode during the follow-up period was the main end point. Twelve hospitals in the research region participated in cholera monitoring.

Additionally, Figure 4 shows that Wilson was in charge of the four-node group that makes up the second biggest team, and Jeronimo, Selma M B was also included. In one of their papers [34], the scientists noted that the epidemiology of visceral leishmaniasis was always changing. Those living in poverty and males were at a higher risk of becoming sick. Late-life diagnoses had increased in frequency in recent years. The endemic nature of visceral leishmaniasis in Natal has resulted in a persistent community of visceral leishmaniasis sufferers. Visceral leishmaniasis, caused by a parasite, continues to be a serious issue for Brazilian public health. There was a positive correlation between L. longipalpis population density and human visceral leishmaniasis.

The top five most prolific writers are shown in Table 3, ranked in order of the number of papers they have contributed in Figure 4. Moreover, the table displays the total number and publication year of articles written by each author.

#### 4.1.2. Institution–Co-Authorship Analysis of Studies of Epidemic Control Based on Big Data Research 

Figure 5 displays the author-like network of intellectual interactions among big data epidemic control research institutes. The size of the nodes represents the number of papers produced by each university, while the thickness of the links represents the level of collaboration between the various universities. 

The institutions that had published four or more articles are designated in Figure 5. Three large, dispersed subnetworks and a few scattered institutions have propped up the whole cooperative network. The Chinese Academy of Sciences worked with many institutions (15 universities), which are not shown in the figure. The University of Oxford created publication links with the Karolinska Institute and worked with up to 11 institutions. Stanford University collaborated with 10 institutions in total, including the University of California, Los Angeles.

To better comprehend Figure 5, Table 4 provides a brief summary of the five most renowned institutions from a variety of nations, ranked according to the number of publications they have published.

#### 4.1.3. Country–Co-Authorship Analysis of Studies on Epidemic Control Based on Big Data Research

Figure 6 displays the co-author networks of big data-related studies within a given country and investigates the global distribution of big data epidemic control research. The number of items from different nations determined the size of the circular nodes. The degree of cooperation between nations is represented by the closeness of their nodes as well as the density of their connections. The larger the purple circle, the more vital the node. The graphic shows three nations with three or more items labelled next to them. 

There were total of eight connections in Figure 6 (having a density of 0.096). The United States and Brazil represent the continent of America, while Japan, India, Bangladesh, and the People’s Republic of China represent Asia; England represents Europe, and Australia is the only representative of the Australian Continent. Except for Romania, Germany, and Australia, the network of cooperation consists of nine relatively key nations that cooperate closely together. The United States, France, and Iran all have purple flags. Thus, these three nations play significant roles in a group of eight nations working together on big data epidemic control studies. 

The top five most productive nations, according by the number of original papers produced, are shown in descending order in Table 5, for convenience in interpreting Figure 6. Of the total number of papers, the United States contributed 83, followed by those from the People’s Republic of China (78), England (35), Australia (26), and Japan (22). 

### 4.2. Co-Citation Analysis of Studies of Epidemic Control Based on Big Data Research

When two or more writers, journals, and articles all appear together in the reference list of a third document, there is a co-citation relationship [35]. It is possible to use the results of such research to map out the fields of study, track the progress of the science, and evaluate the level of overlap between them [36]. Therefore, three primary co-citation analysis forms demonstrate the structure and connections between journal, paper, and author.

#### 4.2.1. Journal Co-Citation Analysis of Studies of Epidemic Control Based on Big Data Research

If an article from two journals appears in the reference section of the same cited article, then it can be shown that both journals are co-cited [37]. Journal co-citation analysis is a useful tool for investigating the structure of the academic sector, whereby scholarly publications serve as the gold standard for dissemination of knowledge [38]. 

Figure 7 is a network diagram in which journals serve as nodes and the connections between them are evidence of their mutual citation. Additionally, the size of the node represents the total number of citations the publication has earned. The journal’s citation frequency is equal to the distance between any two nodes. The larger the node, the more influential the journal, and the closer together the nodes, the more often the journals are cited in each other’s work. 

The primary sub-network, which consists of 393 nodes and accounts for 95% of the whole network, is seen in Figure 7. Most often, the “core journals” refer to the most-cited publications. The counts, main issues, and referenced publications of big data epidemic control research are shown in Table 6, which are necessary for understanding Figure 7.

The possibility of analyzing the referenced journals to discover the core papers and their relationships was explored, and was made possible by the co-citation of publications in big data epidemic control studies. The findings highlighted the multidisciplinary nature of studies using big data and epidemic control. Given the breadth of the disciplines involved in big data epidemic control research, no single area could hope to capture it all.

#### 4.2.2. Document Co-Citation Analysis of Studies of Epidemic Control Based on Big Data Research

The documents were the primary informational resource. This was shown by the occurrence of co-citations. Selecting some exemplary studies to provide an analysis topic for a document’s co-citation link [39] was a crucial step in discovering a domain’s architecture and development route using a big data literature review of a document as a basis for epidemic control. Figure 8 was created in CS, with nodes representing the sources used. The first author and publication year for sources with four or more citations were included. Moreover, the interconnectedness shown by the arrows represents the mutual citation of various sources.

Chinazzi et al., (2020) [40] showed the highest number of citations (around 3385) until 19 January 2023. The authors claimed that the traveling quarantine in Wuhan only slowed the spread of the disease in mainland China by 3–5 days. Up until about the middle of February, case imports were cut by roughly 80%. Using a global population dynamics disease transmission model, the authors predicted how restrictions on international travel could affect the spread of COVID-19 in China. The model was calibrated using worldwide reported cases, and the results reveal that most Chinese towns had already received numerous infected visitors by the time the travel restriction from Wuhan had begun on 23 January 2020.

To better comprehend Figure 8, Table 7 presents the top 12 most cited records together with details such as the number of citations, the year of publication, and the reference.

#### 4.2.3. Author Co-Citation Analysis

Author co-citation analysis were used to identify the study fields of related writers in the subject area and distribute their topics by co-citation network. In addition, it facilitated the identification of key figures in the subject area and the collection of data on the frequency with which their works are mentioned. 

Each node in Figure 9 represents one author, and the line connecting any two writers indicates a co-citation relationship between them. Those who had seven or more references attributed to them are noted in Figure 9. The largest nodes are an unknown individual, an anonymous researcher, a researcher linked with the World Health Organization, and WHO.

To help interpret Figure 9, Table 8 ranks the 15 most cited writers by the total number of citations of their published works. Table 8 shows that Unknown (67) is the most cited author, and that the next fourteen most cited writers are ANONYMOUS (57) WORLD HEALTH ORGANIZATION (28), WHO (26), CHINAZZI M (14), KRAEMER MUG (12), DONG ES (12), TIAN HY (12), BAI Y (11), WU JT (10), ZHU N (8), BONACCORSI G (8), YANG ZF (8), ZHAO S (7), and ALI M (7). These results suggested that these influential authors’ research played critical roles in contributing to big data epidemic control research and its future development. 

### 4.3. Co-Occurrence Analysis of Studies of Epidemic Control Based on Big Data Research

The presence of two or more journals, articles, and authors in a third document’s reference list indicates a co-citation connection [35]. The methodology may be used to map out interdisciplinary pathways, track the growth of scientific knowledge, and quantify the extent to which different fields of study are related to one another [36]. Thus, the structure and connection of journals, documents, and authors are exposed via three primary co-citation analysis types.

#### Keyword Co-Occurrence Analysis of Studies of Epidemic Control Based on Big Data Research

Search terms were used to shed light on the articles’ main ideas. Search phrases (keywords often referenced over time) could indicate frontier themes, and a knowledge map of keyword co-occurrence could indicate hot topics [52]. In Figure 10, which was generated using CS software, each node represents a different term, and its co-occurrence frequency is shown as a line. Figure 10 may be understood by looking at the top 5 most productive keywords in terms of counts, which are shown in Table 9.

### 4.4. Novel Knowledge Clusters

To create the current state of CS research, the best N = 50 slicing technique was selected. If the top N = fifty were used, then the fifty most often occurring terms and references were selected from each section. The reference network for epidemic control with big data is shown in its full bibliographical context in Figure 11. Links between nodes denote sources that were quoted together [10]. Cluster labels are calculated using either the Log-Likelihood Ratio (LLR) or the Mutual Information technique (MI). In this evaluation, LLR was used as the standard, since it consistently produces the best results in terms of a real-world situation after many iterations. Table 10 displays each cluster separately. 

We could tell the size of each cluster in Figure 11 by the number of papers that belonged to it, which in turn represented the intensity of the hotspot. Without a doubt, “COVID-19” was the most cutting-edge field of study. The value of the silhouette shifted from 0 to 1, showing the reliability of each possible cluster arrangement. The silhouette’s relative significance determined how well it clustered. The year in a cluster represents the average year of publication for all works in the cluster. It was possible that the core of a cluster analysis, which was based on the most recent data, represented the patterns that were emerging inside the cluster. Major research frontiers for epidemic control based on big data research are listed in Table 10 as “unified health algorithm, Internet of Things (IoT) platform (2021)”; “COVID-19 infection (2020)” is the major research frontier for epidemic control based on big data research.

#### 4.4.1. Hottest Cluster “COVID-19 Infection”

Infection processing with COVID-19 uses big data epidemic control research as a means of tracking it. Table 10 shows that the largest, group 0 (COVID-19), consists of 31 references with 2020 as the publication year. Because of this, “COVID-19 infection” is an important topic area to focus on while reviewing the relevant literature.

#### 4.4.2. The Emerging Research Topic “Iot Platform and Unified Health Algorithm”

Among the 10 most recent clusters, #7 IoT Platform and Unified Health Algorithm had the most citations (15). The book will not be released until 2021. Cluster 7 exhibits high-quality aggregation, as shown by the silhouette value of 1. The use of big data epidemic control in the search for COVID-19 illness has hit a snag. Big data research approaches for controlling epidemics needs a number of processing methods to be improved to the maximum extent possible so that the innovation can be tracked in a practical manner [53].

The IoT and “unified health algorithm” are two distinct but related concepts. IoT refers to the interconnectedness of devices and systems, allowing for the communication and sharing of data. A unified health algorithm, on the other hand, refers to a specific type of algorithm that can be used to analyze and interpret health data from various sources, such as wearable devices or electronic health records. The combination of these two concepts can lead to the development of an IoT platform that integrates and uses a unified health algorithm to analyze data from connected devices, potentially leading to improved health outcomes and more efficient healthcare delivery [54,55].

### 4.5. Novel Research Frontiers of Big Data Epidemic Control for Future Research

Study hotspots in big data epidemic control are described with the burst keywords, and the research sheds light on emerging trends. After excluding vague terms such as “thing”, which gave no meaning or sense unless joining with other terms, we obtained the top 29 keywords with significant growth between 2011 and 2022 in the big data epidemic control sector (Table 11). Total time is shown by the blue line, while the red line indicates the length of the burst. 

Major research frontiers from 2021 to 2022 with a strength of 1.61 to 1.02 in big data epidemic control research were as follows: i.Influenza: One interesting new area of study is the use of big data to learn more about influenza and how to stop it from spreading. This may be done by combining information from a variety of sources, such as health records, social networks, and search engine data. Machine learning is being utilized to increase the reliability of influenza diagnoses, and forecasting is being developed to predict the virus’s spread. Knowledge of flu and our capacity to prevent and manage epidemics may benefit greatly from the combination of enormous volumes of data and powerful computational methods [56,57].ii.Internet: When it comes to studying and stopping epidemics, the internet is a goldmine of data that can be mined via the examination of internet search data and forms of social media. This information may be used to learn more about the demographics of individuals infected, the spread of the illness geographically, and the nature of the symptoms people are reporting. The internet as a data source for epidemic control is a new area of study that may lead to a deeper knowledge of how epidemics spread and better prevention and treatment methods [58,59].iii.China: By leveraging its growing population and advanced technological infrastructure, China has emerged as a leading research hub in the emerging field of big data control. This field seeks to improve disease diagnosis through the application of machine learning algorithms by analyzing electronic health records and creating predictive models. China is a potential area of investigation in pandemic control because of its dedication to health care and rising efforts with the intention of improving healthcare delivery [60,61,62]. This makes China an ideal place to explore big data-driven answers to various problems presented by infectious illnesses.iv.Provinces: Province specificity is a new area for study that may help us learn about disease transmission in our specific region and create effective measures to curb it. Health systems and other province-specific data sources may shed light on the demographics and lifestyle choices of persons living with the condition, which can in turn be used to pinpoint high-risk locations and drive the design of strategic treatments [63,64,65]. In addition, public health professionals can better react to epidemics in real time with the use of prediction models and machine intelligence. The use of big data to control epidemics in a single province is an exciting new area for study that may lead to substantial advances in our knowledge of disease transmission and the design of more efficient prevention and control measures.v.Human mobility: Human mobility refers to the movement of individuals within and between different locations, and it can be studied using data from GPS, mobile phones, and social media. Big data can be used to understand patterns and trends in human mobility, which can inform public health and transportation policies. For example, big data can be used to identify high-risk areas for the spread of infectious diseases and to optimize the distribution of healthcare resources [66,67,68].vi.Epidemic: Epidemic refers to the rapid spread of a disease within a population, and it can be studied using data from healthcare systems, social media, and other sources [69,70,71]. Big data can be used to detect outbreaks early, track their spread, and identify risk factors. This can help inform public health response and control measures, such as targeted testing, quarantine, and vaccination campaigns [30,46,72].vii.Frameworks: New ground may be broken in the study and management of epidemics with the creation of a complete framework for using big data. Efficient decision-making in the event of an epidemic would be facilitated by the availability of such a framework, which would give an organized approach to the collection, storage, analysis, and interpretation of data from numerous sources [73,74]. Because research techniques could be standardized, findings from various places could be compared with greater ease, and trends could be monitored more closely. More effective and coordinated attempts to control outbreaks may be made with the use of a framework for big data control, which would improve cooperation between scientists and public health professionals.viii.Internet of Things: The Internet of Things refers to the interconnectedness of devices and systems, allowing for the communication and sharing of data [75,76,77]. In the healthcare sector, IoT can enable the collection and analysis of large amounts of health data from various sources, such as wearable devices, electronic health records, and environmental sensors. Big data can be used to analyze and interpret these data, which can lead to improved health outcomes and more efficient healthcare delivery [78,79,80].ix.Air pollution: This refers to the presence of harmful substances in the air that we breathe [81,82]. It can be studied using data from monitoring stations, satellites, and other sources. Big data can be used to identify sources of pollution, track changes over time, and understand the health impacts of air pollution. This can help inform policies and interventions to reduce exposure and improve public health [83,84,85].x.Trend: Big data epidemic control is a relatively new area of study, and the detection and analysis of patterns in the transmission of epidemics using big data is a promising new research horizon in this area. This involves monitoring demographic and behavioral trends in relation to disease transmission, as well as keeping tabs on how often and where outbreaks occur over time [65,86]. Using big data may help public health professionals better understand the dynamics of disease transmission and the causes of epidemics, leading to more efficient measures to curb their spread. Furthermore, trend analysis may aid in the detection of new infectious illnesses and the forecasting of future epidemic trajectories, allowing for quicker and more precise interventions.xi.Systems: A new area of study is the construction of an all-inclusive and quality support for the application of big data to the investigation and management of epidemics. A system such as this would integrate data from EHRs, social media, and wireless sensors to provide a full picture of how diseases are spreading. Real-time monitoring of epidemics and the detection of trends and patterns in pathogens would be possible with the use of modern analysis and modeling techniques. Better decision-making and coordinated control measures would be possible with the use of big data if it were integrated into the current public health systems [76,87,88]. The danger of global pandemics may also be mitigated by the creation of a big data system for epidemic control that would enable a quick and effective response to new infectious illnesses. There is a lot of room for improvement in our capacity to react to and control the spread of infectious illnesses, and this is why developing a complete and integrated system for the use of big data in epidemic management is a promising path for future study.

## 5. Summary and Future Works

Big data has the potential to play a pivotal role in limiting pandemics by facilitating precise forecasting and prompt action. Data collection, storage, and processing may have positive benefits for society and the economy. However, it can also have detrimental effects on the environment, society, and the economy. Cost-benefit analyses of big data initiatives are essential, as are the ethical and responsible use of sensitive data.

The purpose of this bibliometric research was to examine the existing literature on the topic of utilizing large datasets to manage epidemics. CiteSpace (CS) was used to conduct an extensive literature review of the results from a keyword search conducted in Web of Science Core Collections (WOSCC). The study concluded that the United States, the People’s Republic of China, the United Kingdom, Australia, and Japan were the five most prolific countries in terms of the number of original papers published in the area of big data epidemic control research. Authors such as Qadri Firdausi, Fahima Chowdhury, Alejandro Cravioto, Khan Ashraful Islam, and Wilson and groups such as Qadri’s group were singled out for their contributions to this area of research. The Chinese Academy of Sciences was the top intuition, which collaborated with 15 other organizations, and *The Lancet* journal accepted the most papers in this field. Journal co-citation analysis also indicated that some publications received a disproportionately high number of citations from researchers working in big data epidemic control. Use of COVID-19 data, Internet of Things (IoT) platforms, and unified health algorithms were identified as the most promising avenues for further study in this area. “Influenza, internet, China, human mobility, and province” were the emerging keywords in the year 2021–2022, with a strength of 1.61 to 1.2.

## 6. Conclusions

The graphical findings shed light on the intersection of big data and epidemic control, offering new views and insights that may guide future studies, policies, and decisions in this area. Analysis and interpretation of the results were aided by the author’s enhanced familiarity with the data and their context, thanks to the visualizations. By doing so, we were able to better understand the data and identify important trends, patterns, and linkages. The visualized results gave a clear and succinct picture of the data, aiding the reader in grasping the study’s main conclusions and their practical significance. If researchers want to understand more about the connection between big data and epidemic control and how big data may improve decision-making and efficiency during a pandemic, this is a great place to start, as the results were satisfying for both researchers and medicine industry. However, the authors only used the WOSCC to obtain the research papers and analyze them in CS. It is recommended to expand the analysis to other research libraries, such as Scopus, PubMed etc.

## Figures and Tables

**Figure 1 ijerph-20-03930-f001:**
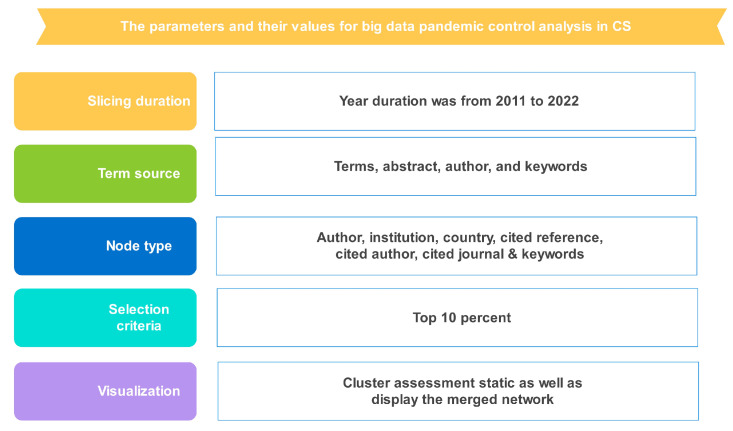
Parameters taken for the big data pandemic control research in CS.

**Figure 2 ijerph-20-03930-f002:**
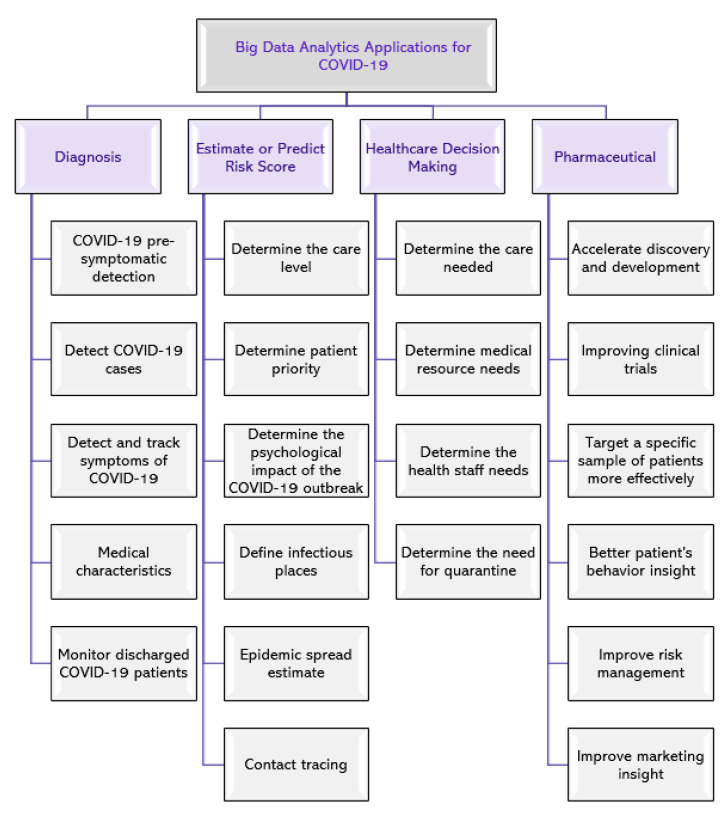
Possible usage of big data analytics in a number of different contexts for COVID-19 [28].

**Figure 3 ijerph-20-03930-f003:**
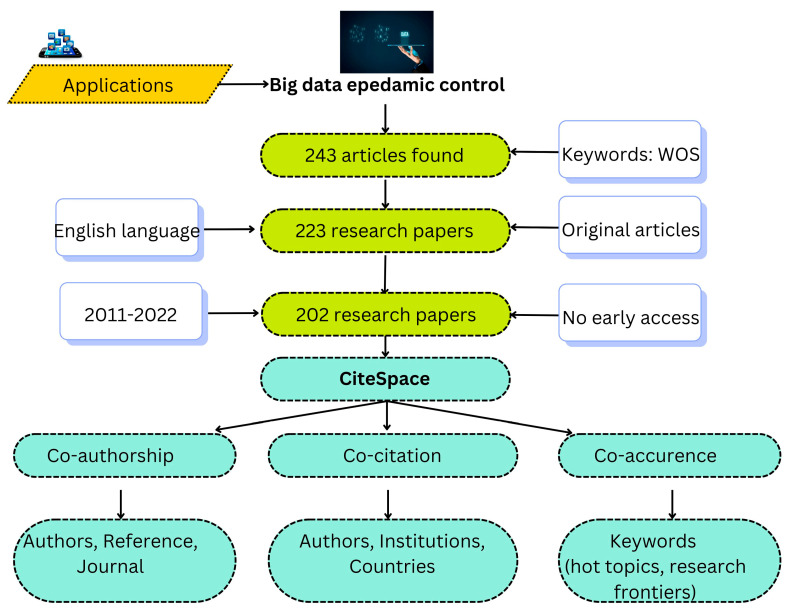
The flowchart displaying the accomplishments for the current study.

**Figure 4 ijerph-20-03930-f004:**
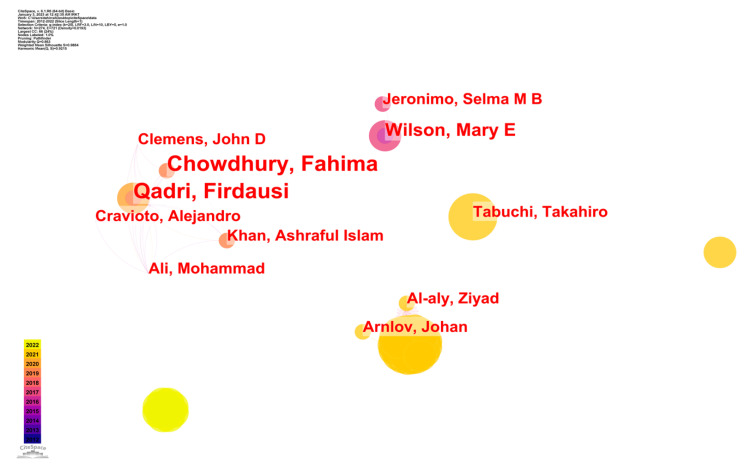
CS Map of Author co-authorship analysis of studies of epidemic control based on big data research.

**Figure 5 ijerph-20-03930-f005:**
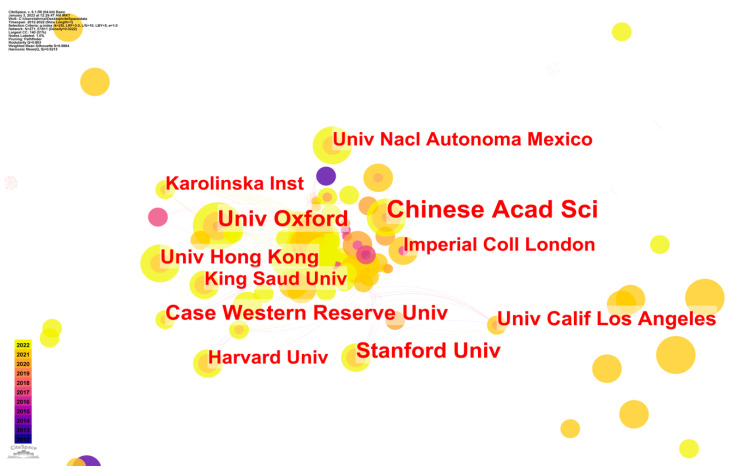
CS map of institution–co-authorship of studies on epidemic control based on big data research.

**Figure 6 ijerph-20-03930-f006:**
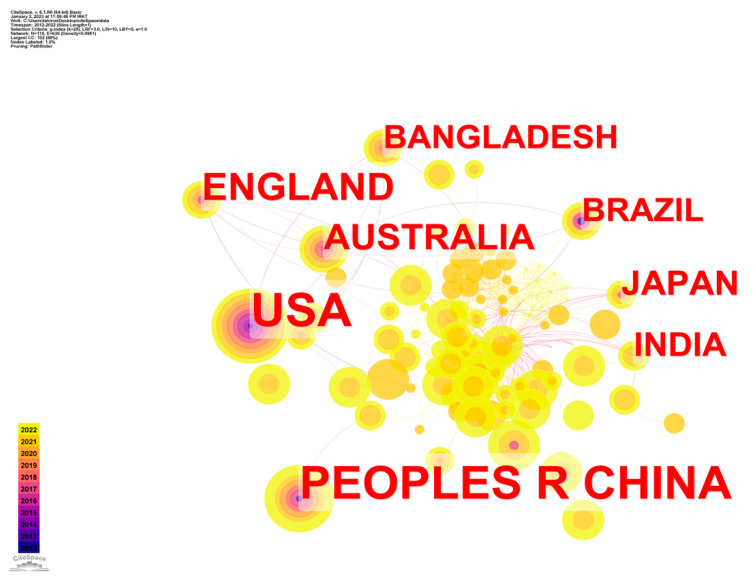
CS map of country–co-authorship studies of epidemic control based on big data research.

**Figure 7 ijerph-20-03930-f007:**
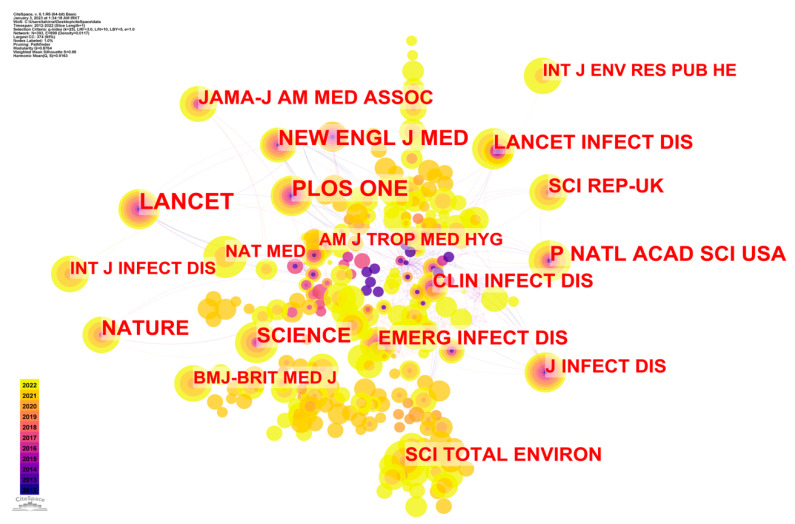
CS map of journal co-citation analysis of studies of epidemic control based on big data research.

**Figure 8 ijerph-20-03930-f008:**
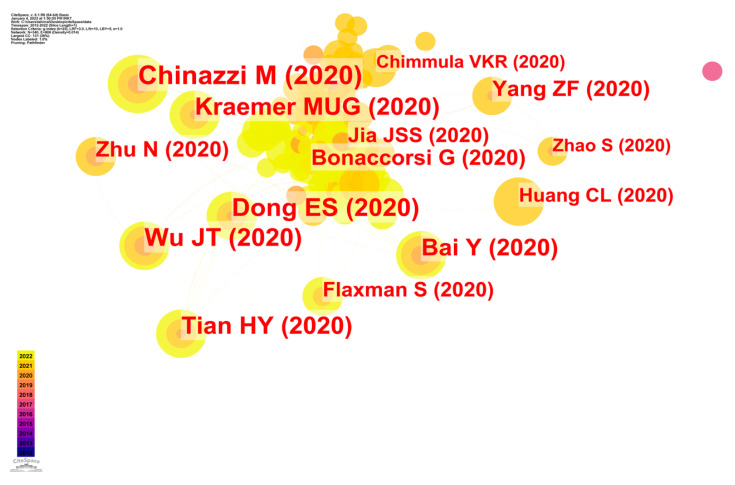
CS map of document co-citation analysis of studies of epidemic control based on big data research.

**Figure 9 ijerph-20-03930-f009:**
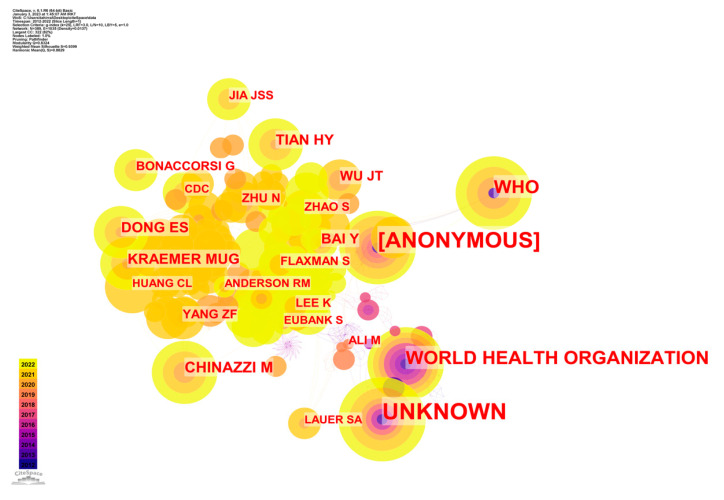
CS map of co-citation author analysis of studies of epidemic control based on big data research.

**Figure 10 ijerph-20-03930-f010:**
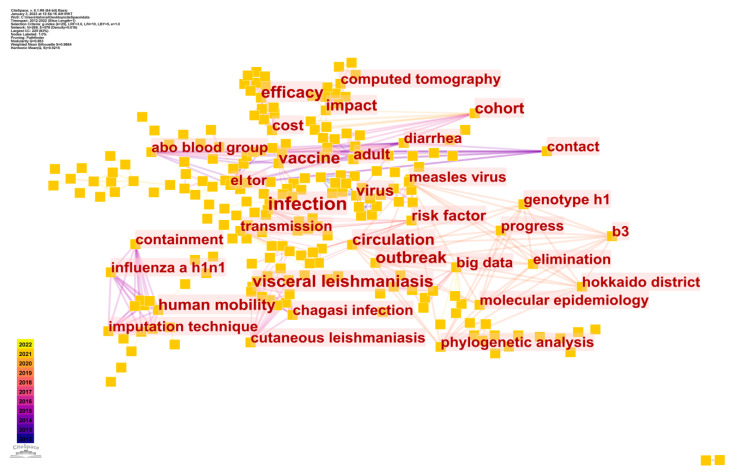
CS map of keyword analysis of studies of epidemic control based on big data research.

**Figure 11 ijerph-20-03930-f011:**
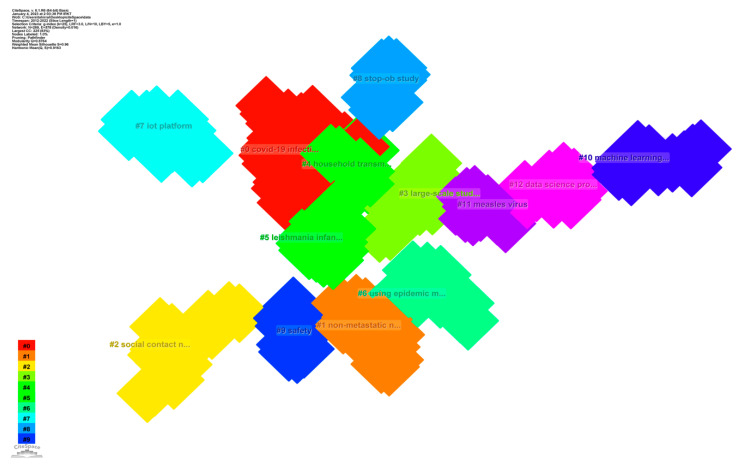
Top 12 co-cited reference clusters of studies of epidemic control based on big data research.

**Table 1 ijerph-20-03930-t001:** Searching codes used and results obtained throughout the data extraction process from WOS.

NO.	Searching Code	Results
1	((“endemic” OR “pandemic”) AND (“city” OR “Urban”) AND (“big data” or “electronic health records” OR “social media posts” OR “sensor data” OR “Artificial intelligence” OR “AI” OR “Machine learning”) AND (“control”))	243
2	After excluding non-articles	223
3	Refined to past 12 years, i.e., 2011–2022	208
4	After excluding early-access articles	202

**Table 2 ijerph-20-03930-t002:** Possible uses of big data for the management of an ongoing COVID-19 outbreak, data accessed from [3].

Duration-Scale	Possible Use	Example
Short-term (weeks)	Identification of a continuing epidemic in a timely manner	The use of big data can make real-time epidemiological data gathering, risk assessment, decision-making processes, and the design and execution of public health initiatives easier to accomplish.
	The diagnosis and outlook for patients with COVID-19	The identification of certain diagnostic and prognostic characteristics is required.
Medium-term (months)	The identification of a potentially applicable therapy strategy	Acknowledgment of certain diagnostic and prognostic aspects
Long-term (decades)	A possible treatment strategy has been identified.	Procurement of medications that are presently on the market while also looking for novel compounds

**Table 3 ijerph-20-03930-t003:** The top 5 most productive authors of studies of epidemic control based on big data research, extracted through CS analysis.

No.	Count	Year	Name of Author
1	6	2014	Qadri, Firdausi
2	6	2013	Chowdhury, Fahima
3	4	2016	Wilson, Mary E
4	3	2017	Arnlov, Johan
5	3	2015	Clemens, John D

**Table 4 ijerph-20-03930-t004:** Top 5 most productive institutions studying epidemic control based on big data research, extracted through CS analysis.

No.	Count	Year	Name of Institution
1	15	2017	Chin. Acad. Sci.
2	11	2020	Univ. Oxford
3	10	2015	Stanford Univ.
4	9	2013	Case Western Reserve Univ.
5	8	2015	Univ. Calif. Los Angeles

**Table 5 ijerph-20-03930-t005:** Top 5 most productive countries studying epidemic control based on big data research, extracted through CS analysis.

No.	Count	Year	Name of Country
1	83	2012	USA
2	78	2013	China
3	35	2013	England
4	26	2013	Australia
5	22	2017	Japan

**Table 6 ijerph-20-03930-t006:** Top 5 most productive journals publishing studies of epidemic control based on big data research, extracted through CS analysis.

No.	Count	Year	Name of Journal
1	91	2012	Lancet
2	79	2012	PLoS ONE
3	67	2012	New Engl. J. Med.
4	60	2013	Proc. Natl. Acad. Sci. USA
5	58	2016	Science

**Table 7 ijerph-20-03930-t007:** Top 12 most cited documents of studies of epidemic control based on big data research, extracted through CS analysis.

No.	Count	Year	Name of Document	Reference
1	14	2020	Chinazzi M, 2020, SCIENCE, V368, P395, DOI 10.1126/science.aba9757	[40]
2	12	2020	Wu JT, 2020, LANCET, V395, P689, DOI 10.1016/S0140-6736(20)30260-9	[41]
3	12	2020	Dong ES, 2020, LANCET INFECT DIS, V20, P533, DOI 10.1016/S1473-3099(20)30120-1	[42]
4	11	2020	Kraemer MUG, 2020, SCIENCE, V368, P493, DOI 10.1126/science.abb4218	[43]
5	11	2020	Tian HY, 2020, SCIENCE, V368, P638, DOI 10.1126/science.abb6105	[44]
6	10	2020	Bai Y, 2020, JAMA-J AM MED ASSOC, V323, P1406, DOI 10.1056/NEJMoa2001316, 10.1001/jama.2020.2565	[45]
7	8	2020	Yang ZF, 2020, J THORAC DIS, V12, P165, DOI 10.21037/jtd.2020.02.64	[46]
8	8	2020	Bonaccorsi G, 2020, P NATL ACAD SCI USA, V117, P15530, DOI 10.1073/pnas.2007658117	[47]
9	8	2020	Zhu N, 2020, NEW ENGL J MED, V382, P727, DOI 10.1056/NEJMoa2001017	[48]
10	7	2020	Jia JSS, 2020, NATURE, V582, P389, DOI 10.1038/s41586-020-2284-y, 10.1109/LGRS.2020.3028443	[49]
11	7	2020	Flaxman S, 2020, NATURE, V584, P257, DOI 10.1038/s41586-020-2405-7	[50]
12	6	2020	Huang CL, 2020, LANCET, V395, P497, DOI 10.1016/S0140-6736(20)30183-5	[51]

**Table 8 ijerph-20-03930-t008:** Top 15 most cited authors of studies of epidemic control based on big data research, extracted through CS analysis.

No.	Count	Year	Name of Cited Author
1	67	2013	Unknown
2	57	2012	[Anonymous]
3	28	2013	World Health Organization
4	26	2013	WHO
5	14	2020	Chinazzi M.
6	12	2020	Kraemer M.U.G.
7	12	2020	Dong E.S.
8	12	2020	Tian H.Y.
9	11	2019	Bai Y.
10	10	2020	Wu J.T.
11	8	2020	Zhu N.
12	8	2021	Bonaccorsi G.
13	8	2020	Yang Z.F.
14	7	2020	Zhao S.
15	7	2013	Ali M.

**Table 9 ijerph-20-03930-t009:** Top 15 most productive keywords in studies of epidemic control based on big data research, extracted through CS analysis.

No.	Count	Year	Keyword
1	16	2016	impact
2	14	2016	infection
3	11	2018	big data
4	11	2015	model
5	10	2018	transmission
6	10	2021	machine learning
7	10	2020	artificial intelligence
8	8	2020	health
9	8	2021	China
10	7	2013	outbreak
11	6	2021	system
12	5	2015	human mobility
13	5	2018	dynamics
14	5	2020	Wuhan
15	5	2014	infectious disease

**Table 10 ijerph-20-03930-t010:** A brief summary of the top 12 clusters.

Cluster-ID	Size	Silhouette	Mean (Year)	Label (LLR)	Label (MI)
0	31	0.908	2020	COVID-19 infection (24.97, 1.0 × 10^−4^)	COVID-19 infection (24.97, 1.0 × 10^−4^)
1	24	0.982	2017	prognostic factor (25.82, 1.0 × 10^−4^)	prognostic factor (25.82, 1.0 × 10^−4^)
2	22	1	2018	high resolution (26.62, 1.0 × 10^4^)	high resolution (26.62, 1.0 × 10^4^)
3	19	0.955	2018	poultry trading network (18.42, 1.0 × 10^−4^)	poultry trading network (18.42, 1.0 × 10^−4^)
4	18	0.89	2016	household transmission (17.98, 1.0 × 10^−4^)	household transmission (17.98, 1.0 × 10^−4^)
5	17	0.976	2016	linkage peak (14.57, 0.001)	linkage peak (14.57, 0.001)
6	16	0.926	2017	policy management (21.14, 1.0 × 10^−4^)	artificial intelligence (0.21)
7	15	1	2021	iot platform (27.11, 1.0 × 10^−4^)	unified health algorithm (0.2)
8	14	0.993	2016	diabetes patient (12.99, 0.001)	balance-related behavior (0.14)
9	13	0.997	2012	component (14.04, 0.001)	state (0.07)
10	13	0.984	2020	Chinese metropolitan city (12.44, 0.001)	national mortality (0.11)
11	12	0.885	2018	measles virus (24.18, 1.0 × 10^−4^)	cross-sectional study (0.13)
12	11	0.924	2018	data science processes (20.87, 1.0 × 10^−4^)	big data (0.07)

**Table 11 ijerph-20-03930-t011:** Top 29 keywords with the deepest citation bursts.

Keywords	Year	Strength	Begin	End	2012–2022
virus	2012	1.84	2012	2013	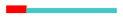
children	2013	1.53	2013	2016	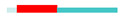
identification	2013	1.25	2013	2013	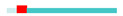
polymerase chain reaction	2013	1.25	2013	2013	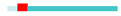
dhaka	2015	1.2	2015	2016	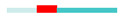
infection	2016	2.33	2016	2020	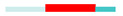
visceral leishmaniasis	2016	2.25	2016	2017	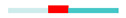
chagasi infection	2016	1.12	2016	2017	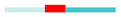
diptera	2017	1.88	2017	2017	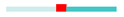
psychodidae	2017	1.25	2017	2017	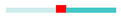
leishmania infantum	2017	1.25	2017	2017	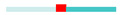
risk factor	2018	1.27	2018	2018	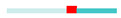
circulation	2018	1.16	2018	2019	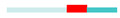
vaccine	2014	1.23	2019	2019	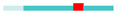
health	2020	1.47	2020	2020	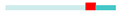
aerosol	2020	1.28	2020	2022	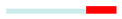
contact tracing	2020	1.23	2020	2020	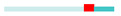
pollution	2020	1.23	2020	2020	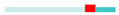
influenza	2021	1.61	2021	2022	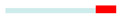
internet	2021	1.61	2021	2022	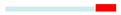
China	2021	1.37	2021	2022	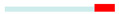
human mobility	2015	1.31	2021	2022	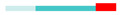
province	2021	1.2	2021	2022	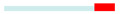
epidemic	2021	1.2	2021	2022	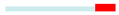
framework	2021	1.2	2021	2022	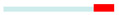
internet of thing	2021	1.2	2021	2022	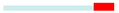
air pollution	2021	1.2	2021	2022	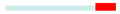
trend	2021	1.2	2021	2022	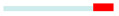
system	2021	1.02	2021	2022	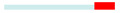

Total time is shown by the blue line, while the red line indicates the length of the burst.

## Data Availability

Not applicable.

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
