# Peer review of "Exploring the Potential of Big Data Analytics in Urban Epidemiology Control: A Comprehensive Study Using CiteSpace"

_ijerph, 2023, doi:10.3390/ijerph20053930_

Round 1

Reviewer 1 Report

1. In the paper, some terms in the text, figures, and tables are inconsistent, which may easily lead to misunderstanding, for example, the United Kingdom (L294) and ENGLAND (Table 5)、LANCET INFECT DIS (Figure 5) and LANCET (Table 6), etc.

2. In several tables (Table 3-9), Centrality is all 0, and the paper lacks more explanations.

Author Response

Authors are grateful to the reviewer for the careful and thorough reading of this manuscript, and for the thoughtful comments and constructive suggestions that helped us to improve the quality of this manuscript. Our response to reviewer comments are as follows (the reviewer’s comments are marked in Red fonts).

Reviewer 2 Report

The paper's topic is interesting and has a good presentation. The result is discussed well: the result of the review of big data and machine learning methods applications in preventing and treating pandemics is clearly presented and illustrated very well for understanding. The abstract reflects the paper's content. The method for study analysis is typical and explained (see Fig. 1). The influence of big data based methods in healthcare is increasing therefore the review of possible methods and studies in epidemiology control is an important problem. This paper provides a review that covers analysis by countries, by years, by types of epidemic control.
According to my point of view, in the paper, issues related to the use of big data in the study of the covid-19 pandemic could be considered more. In particular, many of them were published in MDPI journals:
Levashenko, V.; Rabcan, J.; Zaitseva, E. Reliability Evaluation of the Factors That Influenced COVID-19 Patients’ Condition. Appl. Sci. 2021, 11, 2589. https://doi.org/10.3390/app11062589
Hamid, S.; Bawany, N.Z.; Sodhro, A.H.; Lakhan, A.; Ahmed, S. A Systematic Review and IoMT Based Big Data Framework for COVID-19 Prevention and Detection. Electronics 2022, 11, 2777. https://doi.org/10.3390/electronics11172777
Li, J.; Ma, Y.; Xu, X.; Pei, J.; He, Y. A Study on Epidemic Information Screening, Prevention and Control of Public Opinion Based on Health and Medical Big Data: A Case Study of COVID-19. Int. J. Environ. Res. Public Health 2022, 19, 9819. https://doi.org/10.3390/ijerph19169819
I’d like to recommend extending section 4. Because the COVID pandemic is actual and impacts humans now. Therefore, it is interesting for readers to have a review of methods based on big data analysis which provide decisions on tasks in the fight against coronavirus. Therefore, section 4 should be extended as minimal by the papers introduced above.

Author Response

Authors are grateful to the reviewer for the careful and thorough reading of this manuscript, and for the thoughtful comments and constructive suggestions that helped us to improve the quality of this manuscript. Our response to reviewer comments are as follows (the reviewer’s comments are marked in gray highlights).

Reviewer 3 Report

This study comprehensively surveys existing papers on epidemiological research utilizing big data and visualizes the most contributing research groups, countries, publication and citation trends, which aligns with the theme of the special issue. The search method for the original papers is systematic and the visualized results are  interesting.

A few comments for the authors:

The overall impression is that the manuscript is slightly overly descriptive. The "Summary and Future Works" section also largely taken up by the summary. It would be appropriate for the authors to further discuss their opinions on the visualized results. What value does the author gain from these visualized results and how will the reader receive that value?

Author Response

At the beginning the authors acknowledge the efforts of the reviewer and thank him/her. We are highly pleased that the reviewer has found the work interesting. At the same time, the reviewer has suggested some issues, which the authors believe if corrected would, therefore, increase the value of the article and make it more informative. The article has been modified according to the requirements of the reviewer, and the changed parts are marked with yellow highlights . Moreover, the individual response has been pointed under each issue. Hope that it pleases the reviewer and elevates the scientific value of the article

Reviewer 4 Report

This is an excellent, well-written and well-written study with an original contribution to Exploring the Potential of Big Data Analytics in Controlling Urban Epidemiology: A Comprehensive Study Using CiteSpace.

I encourage its acceptance after the appropriate minor changes as outlined below:

L14: replace <<The purpose of this research was to examine>> with <<The research aimed to examine>>

L149: please replace people' with people's

L 194: please replace people' with people's

L323: please replace was with were

L 342: Top 15 productive documents in epidemic control based on big data research extracted by CS analysis are specified in table 7 title, and 12 documents are presented in the table. Please specify how many were actually used.

L 380: In the title of table 9 is mentioned: <<Table 9 Top 5 productive keywords in Epidemic control based on big data research extracted through CS analysis>>, but in the table are present 15. Please specify how many were actually used.

L 398: “IOT” - please define this acronym

L 407: iot” - please define this acronym

L 408: “IoT” - please define this acronym

L394: the style of the reference list, in its present form, is not in agreement with the journal requirement! Please carefully revise it!

Author Response

At the beginning the authors acknowledge the efforts of the reviewer and thank him/her. We are highly pleased that the reviewer has found the work interesting. At the same time, the reviewer has suggested some issues, which the authors believe if corrected would, therefore, increase the value of the article and make it more informative. The article has been modified according to the requirements of the reviewer, and the changed parts are marked with green highlights. Moreover, the individual response has been pointed under each issue. Hope that it pleases the reviewer and elevates the scientific value of the article.

Reviewer 5 Report

This manuscript requires major revision and I am upset on what is really addressed and discussed by the authors. Most of results come from an application CiteSpace and presentation and structure of the abstract and the article are weird. Therefore it is difficult to understand what are results, interpretation and/or discussion and references are very poor in some sections.

In the abstract, CiteSpace is never mentioned and we don't know what are results, interpretation and/or discussion.

In the introduction, authors focused on megacities because they are "susceptible to the spread of infectious illnesses". They said they want to " give a thorough and up-to-date assessment of the literature on the subject of big data-based urban epidemic control" by "using the bibliometric tool CiteSpace" which is an objective but further they presented 5 research's objectives very different from the primary one and very oriented, probably derived from CiteSpace more than from hypothesis from the authors.

The flowchart in introduction (!?) is disappointing and it lacks the number of paper analyzed and withdrawn at each step. It should be improved and placed in the result section.

The method section is poorly described and we don't know what was really done by authors (except manually exclusion) and how (independent review/selection, agreement between reviewers, kappa ...), and what was done by CiteSpace. Why having focused on WoS only. It is said on CS website : "CiteSpace also provides some simple interfaces for obtaining data from PubMed, arXiv, ADS, and NSF Award Abstracts"

 It is said "CS was used to visualize and evaluate the Epidemic control based on big data research literature". The first oriented objective deal with sustainability of big data and is presented just after the method section. It might be result but it is mainly a discussion on the topic with only 2 references (very poor even as a synthesis for a comprehensive study)

The next paragraph on big data application provide also only one reference to glorify the success of PRC in containing COVID-19.

The fifth point is better exposed but very segmented over 10 pages !! Once again it seems like authors have copied results and figures from CS without a comprehensive synthesis.

In the sixth section on novel research frontiers of big data-epidemic control, it is said some abstract terms were excluded like "infection" but it is presented in the sixth row of the table 11 and it is very unclear how the four major research frontiers were found among those presented in regard with the strength column !?

As a conclusion, authors presented a summary but it lacks global discussion of the results.

In total, it seems like the authors operated with CS and its ability to provide these results without predefined hypothesis or clear objectives a priori and then completed this manuscript with various results proposed by CS without argued discussion. It is reinforced by the short time they have since the end of the study period was only a month ago. Difficult to provide a comprehensive and critical analysis in such a short time !

Author Response

At the beginning the authors acknowledge the efforts of the reviewer and thank him/her. We are highly pleased that the reviewer has given constructive comments which undoubtfully enhanced the quality of paper as well as the authors got chance to correct the mistakes. The article has been modified according to the requirements of the reviewer, and the changed parts are marked with green fonts. Moreover, the individual response has been pointed under each issue. Hope that it pleases the reviewer and elevates the scientific value of the article.

Round 2

Reviewer 1 Report

I think the present version of the article is acceptable.

Reviewer 5 Report

The authors provided relevant responses to all comments and the manuscript was improved. I don't feel qualified enough to judge all the intricacies of English language in the text but I think some turns and passages could be more fluent.

Therefore I decided to accept the revised manuscript with perhaps a review of the English language by a copy editor but editors are better suited to make this request.